# Complement Activation on Endothelial Cell-Derived Microparticles—A Key Determinant for Cardiovascular Risk in Patients with Systemic Lupus Erythematosus?

**DOI:** 10.3390/medicina56100533

**Published:** 2020-10-13

**Authors:** Naomi Martin, Xiaodie Tu, Alicia J. Egan, Cordula Stover

**Affiliations:** 1Faculty of Health and Life Sciences, De Montfort University, Leicester LE1 9BH, UK; xt33@student.le.ac.uk (X.T.); P2414412@my365.dmu.ac.uk (A.J.E.); 2Department of Respiratory Sciences, University of Leicester, Leicester LE1 9HN, UK; cms13@leicester.ac.uk

**Keywords:** lupus, complement, microparticles, cardiovascular

## Abstract

Systemic lupus erythematosus is a classical systemic autoimmune disease that overactivates complement and can affect all organs. Early diagnosis and effective management are important in this immune-complex-mediated chronic inflammatory disease, which has a strong component of vasculitis and carries an increased risk of thrombosis, even in the absence of antiphospholipid antibodies. Development of lupus nephritis can be life limiting but is managed with dialysis and renal transplantation. Therefore, data have become available that cardiovascular risk poses a serious feature of systemic lupus erythematosus that requires monitoring and prospective treatment. Cell-derived microparticles circulate in plasma and thereby intersect the humoral and cellular component of inflammation. They are involved in disease pathophysiology, particularly thrombosis, and represent a known cardiovascular risk. This viewpoint argues that a focus on characteristics of circulating microparticles measured in patients with systemic lupus erythematosus may help to classify certain ethnic groups who are especially at additional risk of experiencing cardiovascular complications.

## 1. Background

Systemic lupus erythematosus (SLE) is an immune-complex-driven disease which engages complement activation via recognition of circulating and bound immune complexes by the classical pathway. In patients, consumption of complement components occurs and its ability to remove immune complexes becomes exhausted. Overactivation of complement in SLE patients is generally noted, leading to the consumption of C3, C4, and a low haemolytic activity test, CH50. A deficiency or low activity of the classical pathway of complement activation predisposes to SLE [1].

Microparticles are inflammatory buddings from cells (between 0.1 and 1 μm) [2], which themselves sustain the inflammatory reaction in autoimmune disease [3]. Microparticles are released from activated, apoptotic or dying cells by a process called blebbing whereby membrane remodelling modification and phosphatidylserine exposure result in an extrusion incorporating parental surface proteins and contents [4]. They are composed of the cell surface proteins and the cytoplasmic contents of their parent cell, are biologically active, and they can act as biomolecular vehicles, transporting proteins and effector molecules to distant sites and affecting target cell function [5]. Microparticles can be isolated from samples (usually plasma or tissue culture supernatants) using serial centrifugation. Flow cytometry is the gold standard method of microparticle analysis [6], whereby microparticles can be enumerated and antibodies to cell surface molecules used for the identification of microparticle phenotype and parental origin.

They are found elevated at significant circulating levels in the blood of patients with SLE [7] and can act as platforms for the assembly of immune complexes and sites of complement activation. Microparticles contribute to the pathogenesis and severity of rheumatic disease, including SLE [8]. Microparticles fuse with target cells and release the contents of their parent cell such as miRNA [9], reactive oxygen species and cytokines [10]. The pathogenesis of the inflammatory response and failure to clear apoptotic cells and debris in SLE could also be further exacerbated by microparticles due to their stimulating and immune complex properties [11].

An indifferent outcome of the extent of complement activation in relation to tissue damage was reported in patients with SLE [12] but immunoglobulin binding subsets of microparticles, which fix complement, are more likely to follow disease activity [13]. Their association with cardiovascular risk in patients with SLE has not yet been studied. A need for studies that stratify severity to novel biomarkers has recently been identified [14].

## 2. Gap in Knowledge

There is an increased incidence of endothelial dysfunction and signs of early atherosclerosis in SLE [15]. Indeed, premature coronary atherosclerosis in SLE constitutes a challenge in the management of patients, the higher risk being due to chronic inflammation and medication [16].

Microparticles are an integral part in the inflammatory reaction of cells and favour a systemic disease. Platelet-derived microparticles generate thrombin, thereby aggravating hypercoagulability. Circulating microparticles, including those derived from endothelial cells, are novel biomarkers in SLE [17], but studies have not related these to clinical disease activity [3,18,19,20] or evaluated for cardiovascular risk in SLE the proportion of IgG-decorated microparticles that also fixed complement [21].

## 3. Endothelial-Derived Microparticles as Markers of Disease Activity in SLE

Endothelial cell-derived microparticles were significantly elevated in patients with SLE [22]. Events positive for Annexin V (microparticle marker) and CD31 (endothelial marker) and negative for CD42b (platelet marker) (Annexin V^+^/CD31^+^/CD42b^−^) were defined as endothelial-derived microparticles. In a longitudinal study, their levels correlated inversely with endothelial dysfunction, measured clinically as flow mediated dilatation of the brachial artery. On follow-up, when the overall clinical score had improved, levels of endothelial cell-derived microparticles were reduced while endothelial dysfunction had normalised [23]. At the same (lower) clinical score, a different study also showed no elevation of endothelial derived microparticles, but evidence of increased microparticles derived from non-lymphoid leukocytes, with markers of apoptosis and elevated glycolytic enzymes [24].

The fraction of endothelial cell-derived microparticles expressing VCAM-1 or C4d was shown to be increased in patients with SLE and vascular disease compared to matched controls [7]. VCAM-1 is a relevant factor because it is a marker of endothelial activation, and its circulating level is considered to predict cardiovascular mortality in SLE [25]; C4d association with microparticles indicates progression of complement activation downstream of C1 binding of bound immunoglobulins. Upon binding of C1, composed of C1q C1r_2_ C1s_2_, to immunoglobulins, a conformational change, autoactivation of dimeric C1r and subsequent activation of dimeric C1s activate the substrate C4. Continuous activation of complement in systemic disease may lead to depletion of individual components; therefore, C4d is used as a biomarker of immune-complex-mediated renal involvement [26].

## 4. Hypothesis Driven Outlook

To gauge the cardiovascular risk of patients with SLE, the carotid intima and media thickness is measured by ultrasound. Many factors have been identified that cooperate in the development of premature atherosclerosis in SLE [27], however, there is diagnostic room for an integrative approach to identify complement activating microparticles at the intersection with the activated endothelium, as novel biomarkers of disease activity. This pursuit, in line with Felten et al.’s third goal (“deriving more comprehensive tools for the evaluation of disease”) [28], may yield revelatory modulators of disease activity that impacts the cardiovascular risk in SLE.

It is important to note that a reduced total copy number of C4 has previously been associated with increased risk to develop SLE [29] and that the C4A allotype (that forms amide bonds with peptides (unlike C4B allotype, which forms ester bonds with carbohydrate surfaces) is important in immune complex induced activation and clearance. C4b is deposited on microparticles [30]. Complement C4 copy number variation has not yet been studied in relation to the inflammatory capacity of microparticles but has been identified as an important determinant in SLE for individuals of Southeast Asian origin [31]. Indeed, complement activation has been in the focus of cardiovascular risk in South Asians for a while [32,33,34].

## 5. A Necessary Focus on Patients with SLE Who Are of South Asian Origin

South Asian individuals have elevated levels of oxidative stress, measured as prostaglandin-F1α oxidative metabolite levels from urine [35], have increased circulating microparticle counts compared to white individuals [36] and are at a disproportionately higher risk of cardiovascular events [37] compared to white individuals; however, the cellular and molecular reasons for this are unclear. South Asian patients suffering from ischemic events showed an increase in circulating endothelial-derived microparticles, but the study did not compare the extent of this increase in another population [38].

South Asian patients with SLE have a higher prevalence of subclinical atherosclerosis and endothelial dysfunction [39] and a higher incidence of thrombotic events [40]. It is known that South Asians develop more severe lupus disease [41]. We think that South Asian patients with SLE constitute a group in whom to test whether complement-fixing IgG containing endothelial-derived microparticles correlate with endothelial inflammation (Figure 1). This is of interest for two reasons: first, to lead on to mechanistic evaluation of the increased risk of cardiovascular complications experienced by South Asians with SLE; second, to stratify a heterogenous population and aid in the clinical management of their disease, where lupus related renal failure remains a leading cause of death [42], and thrombotic risk has not yet been assessed [43]. Incidence of SLE, particularly in Black African Minority Ethnic (BAME) individuals, is more prevalent and possibly underdiagnosed, perhaps due to its complex clinical and serological presentation and relapsing character [44]. These individuals are also at risk of premature cardiovascular complications and infection, and this egregious disparity is of great concern but understudied. Further understanding of the pathogenesis and pathophysiology of lupus disease in ethnic individuals is required.

## Figures and Tables

**Figure 1 medicina-56-00533-f001:**
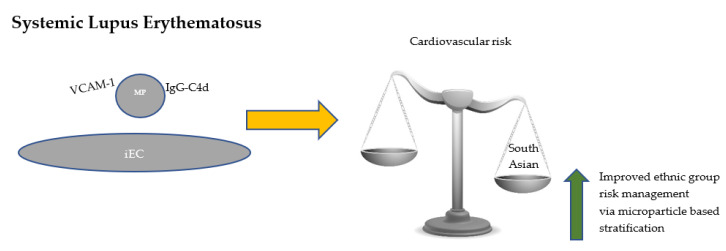
VCAM-1 expressing endothelial derived microparticles that fix immunoglobulins and activate complement circulate in SLE and are pathogenically important mediators of cardiovascular risk, particularly in patients of South Asian origin. Our viewpoint proposes an microparticle-centric analysis of patients to propel understanding of disease and support patient-centred treatment. iEC, inflamed endothelial cell; This Photo (balance) by Unknown Author is licensed under CC BY-SA-NC.

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
