# Peer review of "Complement Activation on Endothelial Cell-Derived Microparticles—A Key Determinant for Cardiovascular Risk in Patients with Systemic Lupus Erythematosus?"

_medicina, 2020, doi:10.3390/medicina56100533_

Round 1

Reviewer 1 Report

In this review, the authors discuss the role of microvesicules in SLE pathogenesis. Although of interest, this review should be improved by discussing in more details different points raised below. 

Line 31, page 1, could the authors better describe how do microparticules can sustain inflammation? could they report the inflammatory molecules trapped in these microparticules or the plasma membrane proteins expressed by these vesicules and responsible for inflammation.

page 2 line 9, a description of the methods/technics used to quantify and isolate blood microparticules is required in this review. In general, a better description of the structure of these microvesicules (membrane composition in term of lipids and proteins and the process of their secretion should be described)

Page 2, line 25; authors have to describe how they can state that the microvesicules are produced by endothelial cells (please could you report the different markers used to reach this conclusion).

Also, authors have to report the molecular links existing between the accumulation of microvesicules in SLE patients and the complement depletion.

Page 2, line 44; could you describe the molecular link between endothelial oxidative stress (is it really endothelial cells or recruited immune cells?) and prostaglandin-F1a oxidative metabolite levels

Author Response

5th October 2020

We thank the Editors for the opportunity to submit our manuscript ‘Complement activation on endothelial cell derived microparticles – key determinant for cardiovascular risk in patients with Systemic Lupus Erythematosus?’ to Medicina for consideration for publication.

We thank the reviewers for their comments and provide detail of our responses below. We have revised our manuscript accordingly and have provided an amended version.

We once again thank the editors of Medicina for considering our manuscript for publication.

With kind regards

Dr Naomi Martin

In this review, the authors discuss the role of microvesicules in SLE pathogenesis. Although of interest, this review should be improved by discussing in more details different points raised below. 

Line 31, page 1, could the authors better describe how do microparticules can sustain inflammation? could they report the inflammatory molecules trapped in these microparticules or the plasma membrane proteins expressed by these vesicules and responsible for inflammation.

We have included information (page 2 line 4) on some components of microparticles which could initiate and sustain inflammation and given information on how this inflammation could be further exacerbated in SLE.

page 2 line 9, a description of the methods/technics used to quantify and isolate blood microparticules is required in this review. In general, a better description of the structure of these microvesicules (membrane composition in term of lipids and proteins and the process of their secretion should be described)

We have included details of the formation (page 1 line 32) and assay (page 1 line 37) of microparticles.

Page 2, line 25; authors have to describe how they can state that the microvesicules are produced by endothelial cells (please could you report the different markers used to reach this conclusion).

We have provided details of the phenotyping of endothelial cell-derived microparticles (page 2 line 25) which were used in the publication (McCarthy et al, 2016).

Also, authors have to report the molecular links existing between the accumulation of microvesicules in SLE patients and the complement depletion

We have corrected and clarified the mechanism of complement activation (page 2 line 37) with additional detail.

Page 2, line 44; could you describe the molecular link between endothelial oxidative stress (is it really endothelial cells or recruited immune cells?) and prostaglandin-F1a oxidative metabolite levels

We have clarified the report of this study (Brady et al, 2012) by deleting the word ‘endothelial’ as the paper present prostaglandin-F1α as marker of systemic oxidative stress, but there  is no significant relationship with ICAM-1. We believe that this modification clarifies the findings of Brady et al.

Reviewer 2 Report

Although the description of this article remains somewhat speculative, this may be interesting for readers of the journal. I have several concerns which should be addressed.

1.     It is needed to draw some figures which summarized the implication of endothelial cell derived microparticles in the pathophysiology of SLE.

2.     Also, to help easily understandings for readers, a table which summarized the core tip of previously literatures in this topic.

3.     The description regarding South Asian patients in its current form is not worth, because the data remains only limited. If the authors would discuss this issue, more relevant data should be added and discussed logically.

Author Response

5th October 2020

We thank the Editors for the opportunity to submit our manuscript ‘Complement activation on endothelial cell derived microparticles – key determinant for cardiovascular risk in patients with Systemic Lupus Erythematosus?’ to Medicina for consideration for publication.

We thank the reviewers for their comments and provide detail of our responses below. We have revised our manuscript accordingly and have provided an amended version.

We once again thank the editors of Medicina for considering our manuscript for publication.

With kind regards

Dr Naomi Martin

Although the description of this article remains somewhat speculative, this may be interesting for readers of the journal. I have several concerns which should be addressed.

It is needed to draw some figures which summarized the implication of endothelial cell derived microparticles in the pathophysiology of SLE.

We have provided a graphical summary of the role of microparticles in the pathophysiology of SLE, particularly in South Asian patients (Figure 1, referred to page 3 line 22).

Also, to help easily understandings for readers, a table which summarized the core tip of previousl literatures in this topic.

We thank the reviewer for this comment. We feel that the scope of our work goes some way to suggest mechanistic mechanisms of exacerbated pathophysiology in South Asian individuals, which has now also been explained further in a schematic diagram. We feel that a table of previous literature would not further clarify these mechanisms for the reader.

The description regarding South Asian patients in its current form is not worth, because the data remains only limited. If the authors would discuss this issue, more relevant data should be added and discussed logically.

We specifically refer to the lack of literature in the area (page 3 line 29). We have provided further information about the possible reasons for underdiagnosis of SLE in South Asian individuals (page 3 line 26) and suggested that further work is required.

Round 2

Reviewer 1 Report

The authors addressed most of my concerns and I recommend this review for publication in Medicina

Reviewer 2 Report

The revised version is almost addressed. I understand this issue.